# Defining the Heart Rate Zone Corresponding to the Lactate Threshold in Colombian Paso Horses

**DOI:** 10.3390/ani15223308

**Published:** 2025-11-17

**Authors:** Angélica María Zuluaga-Cabrera, Guilherme Barbosa da Costa, Iván Darío Martinez, María Patricia Arias

**Affiliations:** 1GISCA Research Group, Faculty of Zootechnics and Veterinary Medicine, Institución Universitaria Visión de las Américas, Medellín 050031, Antioquia, Colombia; 2Bachelor of Veterinary Medicine School, Barretos Educational Foundation University Center—UNIFEB, Barretos 14784-400, SP, Brazil; guilherme.b.costa@unesp.br; 3Faculty of Agricultural and Veterinary Sciences—FCAV, São Paulo State University Júlio de Mesquita Filho—UNESP, Jaboticabal Campus, Jaboticabal 14884-900, SP, Brazil; 4Large Animal Research Group (LARG), Dr. Francisco Maeda College (FAFRAM), Ituverava 14500-000, SP, Brazil; 5Faculty of Agricultural Sciences, Animal Welfare and Ethology Specialization, Fundación Universitaria Agraria de Colombia (UNIAGRARIA), Bogotá 111321, Cundinamarca, Colombia; martinez.ivan1@uniagraria.edu.co; 6INCACES Research Group, Faculty of Veterinary Medicine and Animal Sciences, CES University, Medellín 050021, Antioquia, Colombia; marias@ces.edu.co

**Keywords:** Colombian Paso horses, lactate threshold, heart rate, aerobic training, equine athletes, exercise physiology

## Abstract

Horses used for sport often undertake intense physical exercise, making it important to understand how their bodies respond in order to protect their health and performance. In this study, we explored whether it is possible to estimate a horse’s aerobic capacity using simple measurements of heart rate and blood lactate. By testing horses at different exercise stages, we found that lactate levels increased significantly once the Colombian Paso horses were working above 85% of their maximum heart rate. This predictable change suggests that heart rate and lactate can be used together to estimate aerobic fitness in horses without needing a speed test or invasive procedures. These results provide a practical approach for trainers, veterinarians, and horse owners to better monitor training loads, prevent overexertion, and support the overall welfare of horses.

## 1. Introduction

Colombian Paso horses (CPs) stand out in gait competitions for their endurance, movement precision, and stride frequency, requiring considerable cardiorespiratory fitness to sustain intense performances lasting from 6 to 10 min, during which heart rates range from 160 to 220 bpm [1]. An objective evaluation of these animals’ physical conditioning is essential for guiding training programs and preventing overload.

Among the methods used to assess aerobic fitness in equines, incremental exercise tests have proven effective in identifying physiological effort zones through the analysis of blood lactate thresholds [1,2,3]. The aerobic threshold (LTaer), often associated with lactate concentrations close to 2 mmol/L, represents the transition point between predominantly aerobic metabolism and mild anaerobic activity, serving as a useful reference for training intensity prescriptions [1]. The anaerobic threshold (LTan), typically above 4 mmol/L, marks the intensity at which lactate production exceeds clearance, indicating the onset of metabolic fatigue [3].

Various approaches have been proposed to estimate these thresholds, including visual methods, segmented regression models, and curve-based lactate analyses. The gold standard, however, remains the determination of the maximal lactate steady state (MLSS) in horses and humans [4,5]. In parallel, the use of only heart rate as a non-invasive marker has gained importance, particularly under field conditions, due to its practicality and its strong correlation with metabolic variables—especially in horses for which stride frequency, rather than speed, is the key determinant, such as in gaited breeds like Colombian Paso horses (CPs). Wearable devices, such as heart rate monitors and GPS units, can be fitted to horses non-invasively and comfortably [6].

The classic approach in sports medicine establishes a direct relationship between heart rate zones and lactate production. However, in Colombian Paso horses (CPs), some studies have indicated that this breed may produce lactate at very early stages, even when the heart rate corresponds to what is typically considered a low-intensity zone [7]. Three threshold estimation methods—visual inspection, the fixed 2 mmol/L (ZL2), and the 4 mmol/L (ZL4)—were selected to compare traditional fixed-point criteria with a visual approach that considers breed-specific metabolic patterns. This combination allows evaluation of both standardized and adaptive interpretations of lactate kinetics in Colombian Paso horses, whose locomotor pattern differs from that of high-speed breeds.

Nevertheless, few studies have validating the relationship between heart rate zones and lactate thresholds in Colombian Paso horses (CPs)—a breed whose gait and metabolic characteristics differ markedly from those of other breeds, such as Thoroughbreds and Arabians (measured by oxygen consumption) [8,9]. We hypothesize that mathematical models could be applied to predict the predominant metabolic pathway during exercise in CPs under field conditions.

Therefore, this study aimed to determine the correspondence between heart rate zones and lactate thresholds in Colombian Paso horses using three predictive methods.

## 2. Materials and Methods

The experiment was conducted in a “Very Humid Lower Montane Forest” life zone, located at 2130 m above sea level, with ambient temperatures ranging from 12 to 18 °C and relative humidity of 69%. The study followed ethical guidelines and was approved by the Animal Ethics Committee of the University of Antioquia under protocol #122 on 5 February 2019.

Eighteen Colombian Paso horses (CPs), both male and female, with an average weight of 371 ± 30 kg were selected. The animals were under complete housing and fed on pangola grass hay (95% dry matter *Digitaria eriantha*; 2.5 kg/d on average), green forage (20% dry matter *Pennisetum purpureum*; 30 kg/d on average), commercial balanced feed (2 kg/d on average), mineral salt (100 g/d), and water ad libitum. Before the sample phase, all animals underwent physical, hematological, and biochemical examinations to ensure a healthy status.

### 2.1. Groups

Nine untrained horses (mean age: 38 ± 6.2 months; males = 4, females = 5) and nine trained horses (mean age: 83 ± 26.9 months; males = 4, females = 5) were conveniently selected. The animals were assigned to two experimental groups: the untrained group (GD), consisting of horses at the beginning of their performance process, and the trained group (GT), consisting of horses undergoing a continuous and stable training program.

### 2.2. Incremental Stress Testing (IET)

The selected animals underwent a standardized field exercise test lasting approximately 30 min, consisting of four progressive intensity stages, with rest periods and a final recovery phase [9]. Heart rate (HR) was monitored using an Ambit 3 sensor (Suunto^®^, Vantaa, Finland). The HR monitor had been validated by comparison with another device and with an electrocardiograph [10]. Exercise intensity was defined according to HR zones (Figure 1): warm-up/zone 1 (55–65% HRmax), moderate/zone 2 (65–75% HRmax), high/zone 3 (75–85% HRmax), and maximal/zone 5 (≥85% HRmax). Each stage included a 1-min rest interval (IET), a fixed maximal heart rate of 220 bpm was used according with a previous study [10]. The track measured 35 m in length and 20 m in width, covered with dry sand similar to that used in competitions, and the exercise was performed in an oval pattern. Each stage consisted of 5 min of continuous exercise [11] (Figure 1). During the IET, intensity was controlled by maintaining consistent gait cadence typical of competition to elicit the heart rate responses in each performance stage, guided by an experienced rider (using bride and leg movements), while heart rate was continuously recorded for subsequent analysis. This approach ensured that HR corresponded to the heart rate zone required.

**Figure 1 animals-15-03308-f001:**
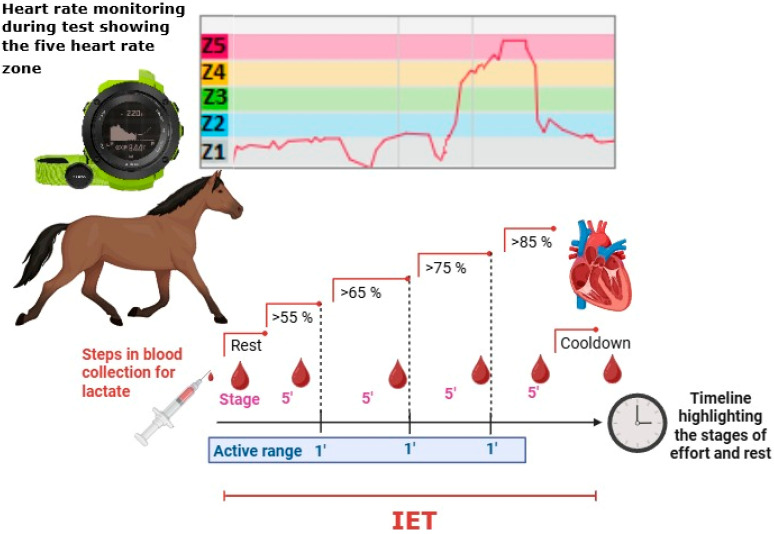
A diagram representing the respective test performed. At the top, the Ambit 3 heart rate monitor shows the heart rate zones in relation to the test stages. At the bottom, the 5-min exercise periods, the 1-min active intervals, and all points where blood samples were collected for lactate measurement are highlighted. Blood drop represents the sample collection moments; 1′ represents active rest between stages; 5′ represents effort stages performed at the respective heart rates; bpm: heart-beats per minute; IET: Incremental exercise test.

### 2.3. Blood Collection

Blood samples for lactate analysis were collected using a Nova Plus device (Nova Biomedical, Waltham, MA, USA) before the IET (rest), at the end of each IET stage, and after the IET (cool-down). Lactometers were previously validated for equine measures [12]. Approximately 4 mL of blood was collected via jugular venipuncture using vacuum tubes with 25 × 0.8 mm needles and EDTA + sodium fluoride anticoagulant. Additional samples were taken for CK (creatin-kinase), creatinine, BUN (blood ureic nitrogen), and AST (aspartate amino transferase) analysis before (at rest) and after exercise (at the end of IET), placed in dry tubes, and analyzed using enzymatic kinetic colorimetric methods.

### 2.4. Determination of the Heart Rate Zone Corresponding to the Lactate Threshold

The results from each horse’s lactate samples, according to each HR Zone, were analyzed using Excel, following the methods of Ferraz et al. [13]. All methods are illustrated in Figure 2 and described below.

ZL4 (red line) represents the frequency zone in which lactate corresponds to 4 mM; ZL2 (blue line) represents the frequency zone in which lactate corresponds to 2 mM; the purple arrow represents the frequency zone in which lactate corresponds to visual analysis.

Pre-established values for ZL2 and ZL4 in horses were used; these points correspond to the lactate concentration at which several authors have reported a shift from linear to exponential lactate production, surpassing clearance rates [3].

The visual method consisted of identifying the point at which the lactate concentration increased non-linearly, representing an exponential rise [14]. This analysis was performed by two experts in the methodology, who defined the point corresponding to the visual lactate threshold (Figure 2).

Once the thresholds were estimated, they were correlated with heart rate zones: Zone 1 (<55% HRmax), Zone 2 (55–65%), Zone 3 (65–75%), Zone 4 (75–85%), and Zone 5 (≥85%). The maximum heart rate was estimated based on literature, which indicates that CP horses have a maximum HR near 220 bpm. The heart rate zones were calculated accordingly [2,3,10].

### 2.5. Experimental Design and Statistical Analyses

Statistical analyses were performed using SigmaPlot software version 14.5 (Systat Software Inc., San Jose, CA, USA). Data were tested for normality (Shapiro–Wilk) and homogeneity of variance (Levene’s test). For lactate and biochemical variables, a mixed-effects model was fitted with group (trained vs. untrained), time (before, during, and after exercise phases), and their interaction as fixed effects, and animal as a random effect to account for repeated measures within every individual.

The repeated-measures covariance structure was modeled using compound symmetry, selected based on the lowest Akaike Information Criterion (AIC). When significant effects were detected, Tukey’s post hoc test was applied.

Because heart rate zones are ordinal categories (not continuous variables), comparisons between zones were performed using the nonparametric Friedman test for repeated measures within groups, followed by Dunn’s multiple comparisons test. The significance level adopted was *p* ≤ 0.05.

## 3. Results

This study provides a systematic analysis of the feasibility of estimating the aerobic lactate threshold (LTaer) in horses using predictive methods based on blood lactate levels. There was no significant difference between groups when analyzing lactate values obtained during the test. Regarding heart rate, similar statistical patterns to those of lactate were observed, with no differences between groups, but significant within-group differences between the >85% (zone 5) and 75–85% (zone 4) HRmax stages and the other stages. These results are shown in Figure 3.

Based on the blood lactate data, predictive methods were applied to identify the aerobic threshold and, from this point, to determine the corresponding heart rate zone. The methods were applied to both groups (Table 1), identifying the heart rate zone in which each threshold fell. Both the trained (GT) and untrained (GD) groups exhibited identical results, with the ZL4 method placing the threshold at heart rate zone 4. The ZL2 threshold corresponded to zone 2 for GT and zone 3 for GD. Using the visual method, the average threshold was around 1.32 mmol/L, corresponding to heart rate zone 2.

To assess the physiological changes induced by exercise, biochemical parameters such as CK, AST, creatinine, and urea (BUN) were analyzed, aiming to demonstrate systemic changes resulting from the imposed exercise load. The results of these analyses are shown in Figure 4.

## 4. Discussion

Despite the popularity of Colombian Paso horses (CPs) in Colombia and other countries, there is a lack of studies addressing their physical work capacity, and training programs designed for this breed often lack scientific foundation, limiting their maximum athletic potential. Predictive methods for determining the aerobic lactate threshold (LTaer) have already been applied to human athletes in various sports [15,16,17,18,19], as well as to dogs [13], rats [15], and horses [3,5,20,21].

For many years, measuring physiological parameters such as heart rate percentage and blood lactate concentration during and after exercise has been a widely used method to assess physical effort intensity [2,22,23]. In CPs, these parameters, combined with lactate threshold predictors, are highly useful for guiding training loads, especially considering the limitations of field-based testing. The predictive approach used in this study offers a relevant opportunity to deepen the understanding of the relationship between exercise intensity and metabolism in CPs.

This study proposes an additional method for predicting LTaer under field conditions in CP, based on visual assessment of the onset of the lactate increase (slope) and heart rate. Among several methods available for estimating LTaer using the speed–lactate curve [5,23], the main conclusion of the present study is that, under field conditions, it is feasible to estimate the exercise intensity corresponding to LTaer in CPs using some of the tested heart rate–based methods. In a more refined and validated model, this approach could eventually eliminate the need for blood sampling by non-medical personnel such as trainers or riders, enabling them to monitor training intensity and prevent exertional injuries related to overtraining using only heart rate monitors.

Identifying the lactate threshold is a well-established and essential parameter for evaluating athletic potential and guiding training programs [22,24,25,26,27]. In horses, this threshold is a crucial predictor of exercise intensity and fitness level in endurance training [28]. Heart rate data are also widely used to monitor and analyze physical training in athletes [6,22,29]. Therefore, establishing the LTaer in CPs and training them based on the heart rate zones where this threshold occurs may represent a more accurate approach, contributing to improved physical fitness and, most importantly, reducing the risk of musculoskeletal injuries and other overload-related conditions.

In this study, most horses presented LTaer within zone 2 according to the visual method for both groups, zone 2 for the trained group (GT), and zone 3 for the untrained group (GD) using the ZL2 method, and zone 4 using the ZL4 method. These findings indicate that if the training goal is to improve aerobic capacity, exercises should be performed within these intensity zones. This is feasible and can be effectively controlled using portable heart rate monitors. This finding is especially important considering that riders often train CPs at heart rates above 75% of HRmax (zone 5). Current literature reinforces the necessity of training in different heart rate zones to develop both aerobic and anaerobic excellence in equine athletes [5,29].

A fixed HRmax of 220 bpm was used based on prior studies that documented consistent maximum heart rate values across gaited Colombian breeds during incremental exercise tests under similar field conditions [7,10]. While interindividual variation is recognized, this fixed reference allows standardized comparison across horses when breed specific HRmax ranges are narrow.

In humans, studies have already demonstrated significant variations in blood lactate concentrations among individuals [30] and across different sports disciplines [31]. Since equine exercise physiology concepts were largely adapted from human medicine, it is relevant to investigate these indices in different equestrian modalities. In Thoroughbred horses, for instance, it has been shown that V4 does not adequately represent the lactate threshold [32]. In Arabian horses, the maximum lactate steady state (MLSS) concentrations were significantly lower than 4 mmol/L in various treadmill protocols, indicating that V4 is also not suitable for predicting this threshold in that breed [33]. Therefore, practical methods such as those proposed in this study may represent a significant advancement in CP training. Structuring training based on heart rate zones could be a crucial step in enhancing the athletic development of this breed.

The relatively low visual threshold (~1.32 mmol/L) may reflect breed-specific adaptations in Colombian Paso horses, where sustained rhythmic gaits rely on efficient oxidative metabolism and early lactate clearance in some of the gait variants of the breed (trot/trocha and gallop). Similar low thresholds have been described in other CP studies related to muscle metabolism and specific-breed gait variations [7].

From a practical standpoint, the ZL2 method seems to offer the best compromise for prescribing workload intensity in Colombian Paso horses, as supported by the statistical differences observed between groups. However, although training CPs using the ZL2 threshold ensures predominantly aerobic exercise, it remains essential to adjust the workload periodically to prevent performance stagnation or performance plateau.

The absence of statistically significant differences in blood lactate concentration between trained and untrained groups was not unexpected. Similar findings have been described in other equine breeds when training status was moderate or when both groups performed submaximal workloads [34]. In CPs, the gait mechanics and aerobic predominance of the discipline likely reduce the magnitude of inter-group metabolic differences. Practically, this suggests that lactate accumulation alone may not be a sensitive discriminator of training level in gaited breeds, as small changes in muscular oxidative capacity could occur without altering systemic lactatemia.

Intra-group differences were expected because the lactate concentration increases exponentially once the workload surpasses the anaerobic threshold, whereas inter-group variation depends on long-term metabolic adaptations. Horses in both groups likely reached similar relative exercise intensities; however, trained horses may have delayed lactate accumulation onset rather than reducing its absolute peak. This pattern is consistent with physiological adaptations described in endurance horses, where improved mitochondrial efficiency shifts the lactate curve to the right [35].

The similarity between HR and lactate responses reinforces their strong correlation in equine exercise physiology [3]. Although mean HR values did not differ significantly between trained and untrained horses, both groups exhibited expected increases in HR at higher effort levels (>85% HRmax). The absence of inter-group differences suggests that training-induced cardiac adaptations in CPs may not manifest as changes in maximal HR but rather as enhanced recovery rate or reduced HR variability post-exercise.

Among the threshold methods, the ZL2 criterion was the only one able to distinguish between groups, placing trained horses in a lower HR zone than untrained ones. This supports its potential practical value for field-based training monitoring. The ZL2 threshold (2 mmol/L) is frequently associated with aerobic capacity and has been validated as a reliable predictor of endurance performance in several breeds [19,28,31]. Thus, using ZL2 to estimate the lactate threshold may allow trainers to tailor workload intensity and avoid premature anaerobic engagement during gait training.

The post-exercise increase in biochemical markers such as CK, AST, creatinine, and BUN was expected and reflects transient muscular and metabolic stress associated with physical exertion. The magnitude of change was moderate, consistent with reversible physiological responses rather than tissue damage [1]. These markers indirectly support the HR and lactate findings, indicating that the exercise intensity was sufficient to challenge metabolic homeostasis without inducing pathological alterations. Previous studies in CPs have reported comparable patterns, with CK and AST elevations correlating with exercise duration and intensity [1]. No significant differences were found between experimental groups, indicating that the workload was evenly distributed among all animals.

Overall, the obtained results suggest that lactate and HR dynamics in CPs reflect a predominantly aerobic metabolism during gait performance, even at intensities above 85% HRmax. This supports the feasibility of using HR-based models for estimating lactate thresholds under field conditions, particularly through the ZL2 criterion. Nevertheless, the application of a fixed HRmax (220 bpm) represents a limitation, as individual HRmax variability may affect precision in zone assignment. Future research should consider individualized HRmax determination and the inclusion of recovery indices to refine these models.

The limitations of this study include the small sample size and the use of mixed-gender groups; therefore, the results should be interpreted with caution and in accordance with the study context. An important limitation of this approach is the use of a generalized HRmax value, which may underestimate individual variability and slightly affect precision when prescribing training intensity. Nevertheless, this assumption provides a standardized framework applicable for population-level training analysis.

Establishing the MLSS in CPs would be an important step toward developing training protocols more precisely tailored to the physical capacity of these horses. Future studies should focus on expanding the use of heart rate zones in CP training, including identifying the zone corresponding to the MLSS, especially considering the accessibility of wearable technology and its potential positive impact on performance.

## 5. Conclusions

In conclusion, the findings indicate that heart rate zones, when coupled with threshold-based modeling, can serve as a practical tool for estimating aerobic performance in gaited horses. These data may be applied to optimize training load, monitor conditioning progress, and prevent exertional overtraining. However, further studies are warranted to establish individualized HRmax values and validate these models across different gaits and training regimens.

## Figures and Tables

**Figure 2 animals-15-03308-f002:**
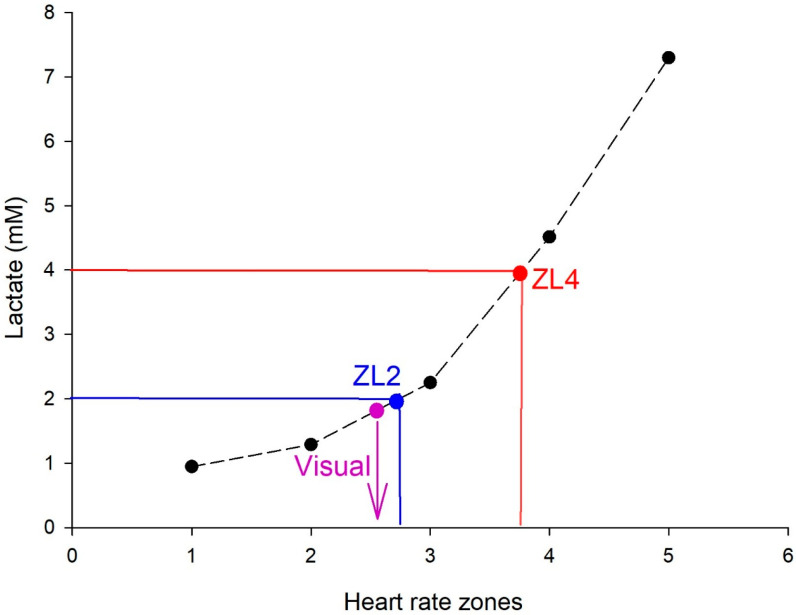
Demonstration of how the pre-fixed and visual methods will be applied in relation to the curve generated by the test and the Heart Rate Zones corresponding to the Lactate Threshold.

**Figure 3 animals-15-03308-f003:**
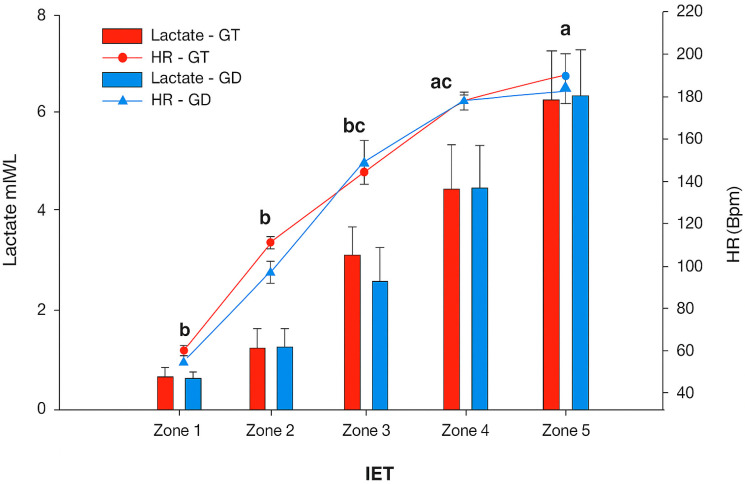
Means and standard errors of blood lactate concentrations and heart rate of trained and untrained CPs during the execution of the IET. HR: heart rate; BPM: beats per minute; Lowercase letters represent statistical difference over time. Differences were found in all intragroup parameters over time (*p* ≤ 0.05). No statistical differences were found between groups.

**Figure 4 animals-15-03308-f004:**
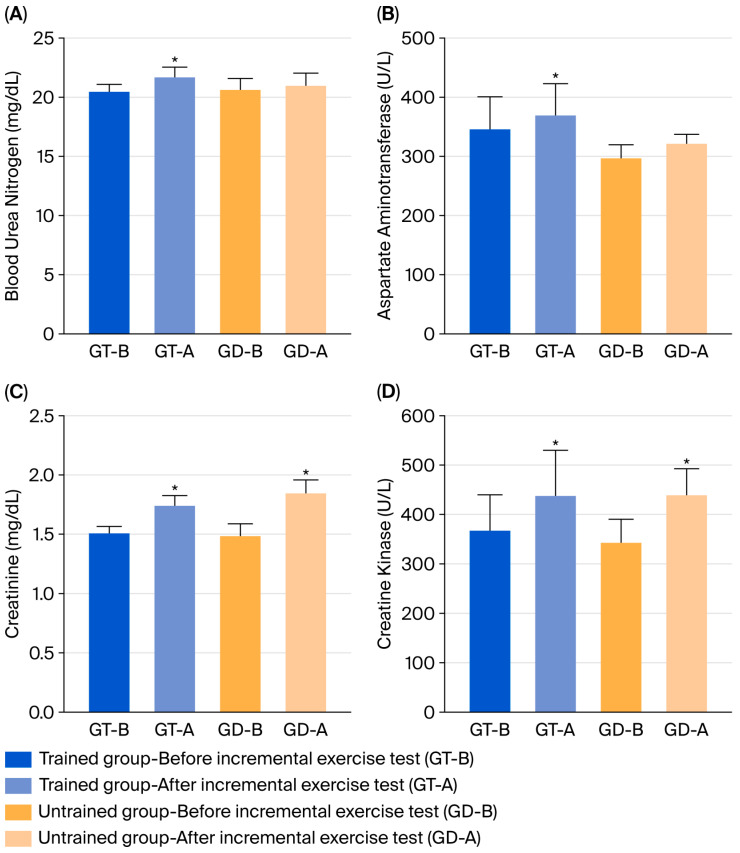
Biochemical changes in trained and untrained CPs before and after an IET. GT/GD-B = before IET; GT/GB-A = after IET. * Statistical difference *p* ≤ 0.05. (**A**) Blood ureic nitrogen concentration in trained and untrained CP horses, before and after IET; (**B**) Aspartate aminotransferase blood concentration in trained and untrained CP horse, before and after IET; (**C**) creatinine blood concentration in trained and untrained CP horse, before and after IET; (**D**) creatinine kinase blood concentration in trained and untrained CP horse, before and after IET.

**Table 1 animals-15-03308-t001:** Heart rate zones associated with each method for GT and GD animals. Mean and standard error of heart rates are also described.

	GT		GD	
	HR Frequency Zone	Heart Rate (bpm)	HR Frequency Zone	Heart Rate (bpm)
ZL2	3	145 ± 5	3	149 ± 7
ZL4	4	175 ± 4	4	176 ± 4
Visual	2	107 ± 6	2	93 ± 8
*p*	≤0.01	≤0.01	≤0.01	≤0.01

ZL4: frequency zone in which lactate corresponds to 4 mM/L; ZL2: frequency zone in which lactate corresponds to 2 mM/L; GT: trained group; GD: untrained group; *p*: statistical difference between the methods within the same group. In heart rate, differences were also found between the zones. No statistical differences were found between the groups.

## Data Availability

Data available upon request.

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
