# Peer review of "Defining the Heart Rate Zone Corresponding to the Lactate Threshold in Colombian Paso Horses"

_animals, 2025, doi:10.3390/ani15223308_

Round 1
Reviewer 1 Report
Comments and Suggestions for Authors
Dear editors,
In the present study, the authors estimate the aerobic lactate threshold (LTaer) using non-invasive methods and correlate it with heart rate (HR) zones. The rationale of this research is the need to provide practical and non-invasive approaches to monitor training loads, prevent overexertion, and support the welfare of horses. The references are proper. Some important information is missing in the “Materials and Methods” section. Therefore, I have requested that the authors revise the manuscript to ensure its scientific soundness. In my opinion, this manuscript, since its topic is interesting and it provides useful information in the field of equine exercise physiology, can be considered for publication after minor revisions.
The Summary and Abstract give a clear and concise overview of the main text without unnecessary repetition. The Summary includes all the elements required by the journal’s guidelines. However, the Abstract needs revision, as it is currently too long and should be shortened. Please note that, according to the journal’s rules, the Abstract must not exceed 200 words.
The Title accurately reflects the content of the paper. The Keywords are appropriate and effectively describe the topic. The Introduction offers detailed information on the current state of knowledge in this field and successfully contextualizes its main aspects, which is helpful for readers who may not be familiar with the subject. However, this section would benefit from a clearer explanation of why predicting the aerobic lactate threshold (LTaer) is important for the horse’s management and performance. Although the aim of this study is stated, the hypothesis is missing. Please include your hypothesis before the aim, as it represents the rationale that motivated your research.
Materials and Methods are well-developed and presented. However, some important information necessary for a physiological study is missing. The number of females and males used should be expressed and also their number in the two groups (untrained and trained). Could you provide more detailed information about the enrolled animals, such as their diet and farm management? Please, integrate all this information into the text. Figures 1 and 2: the figures are proper and add useful information about the design experiment; however, they should be enlarged and the caption should be contain a more detailed description about the images and plots. Figure 3: Please enlarge the figures and arrange them side by side to reduce space and make the manuscript easier to read. Please, in the main text, explain acronyms such as CK, AST, BUN (Blood Urea Nitrogen). The manuscript should clarify the statistical assumptions underlying the analyses. Specifically, it should indicate whether data normality was assessed before being applied to ensure that the use of parametric tests is justified. The p-value should be written with the ≤ symbol. Fix it in the text and caption where it appears.
The Results are organized coherently and illustrated with clear plots and tables. The figure 4 is visually clearly structured; however, it should be enlarged because it is too small.
The Discussion and Conclusion are clearly written and well structured, providing persuasive interpretations. However, certain sections would benefit from revision. In the Discussion, could the authors better contextualize how these findings may impact current training practices? Please also include the limitations of your study, such as the small sample size and groups of mixed gender. The discussion mentions that most horses had their LTaer in zone 2 or 3. Could the authors provide a clearer interpretation of what this means practically for trainers (e.g., how to structure daily sessions based on these findings)?
In my opinion, since the topic is interesting and the manuscript provides useful information in the field of equine exercise physiology, it can be considered for publication after minor revisions.
Author Response
Thank you very much for taking the time to review this manuscript. Please find the detailed responses below and the corresponding revisions/corrections highlighted in the re-submitted files.
Comment 1: [the Abstract needs revision, as it is currently too long and should be shortened. Please note that, according to the journal’s rules, the Abstract must not exceed 200 words.]
Response 1: Thank you for pointing this out. The abstract was revised by the editorial author services to guarantee journa´s rules.
Comment 2: [this section (introduction) would benefit from a clearer explanation of why predicting the aerobic lactate threshold (LTaer) is important for the horse’s management and performance. Although the aim of this study is stated, the hypothesis is missing.]
Response 2: According with your comments we modified the text, please see the changes starting in the line 94, in the introduction section.
Comment 3: [The number of females and males used should be expressed and also their number in the two groups (untrained and trained).]
Response 3: Done. Please see the number of males and females by group in the manuscript, line 115.
Comment 4: [Could you provide more detailed information about the enrolled animals, such as their diet and farm management?]
Response 4: The information was added in the text. Please find it in line 108.
Comment 5: [Figures 1 and 2: the figures are proper and add useful information about the design experiment; however, they should be enlarged and the caption should be contain a more detailed description about the images and plots. Figure 3: Please enlarge the figures and arrange them side by side to reduce space and make the manuscript easier to read. Please, in the main text, explain acronyms such as CK, AST, BUN (Blood Urea Nitrogen)]
Response 5: Changes in the figures were made by editing author services. Acronyms were explained in line 136 and 158.
Comment 6: [The manuscript should clarify the statistical assumptions underlying the analyses. Specifically, it should indicate whether data normality was assessed before being applied to ensure that the use of parametric tests is justified. The p-value should be written with the ≤ symbol. Fix it in the text and caption where it appears.]
Response 6: The statistical analysis was re-written, please find it in line 178. Symbol of P-value was changed.
Comment 7: [The figure 4 is visually clearly structured; however, it should be enlarged because it is too small.]
Response 7: The figure was enlarged, however the editorial author services will improve it.
Comment 8: [In the Discussion, could the authors better contextualize how these findings may impact current training practices?] and [he discussion mentions that most horses had their LTaer in zone 2 or 3. Could the authors provide a clearer interpretation of what this means practically for trainers (e.g., how to structure daily sessions based on these findings)?]
Response 8: Agree. We added a text in one paragraph, please see it in the line 322.
Comment 9: [Please also include the limitations of your study, such as the small sample size and groups of mixed gender]
Response 9: Done. Limitations were included, you can find it in the line 342.
Reviewer 2 Report
Comments and Suggestions for Authors
Comments and Suggestions for Authors
Thank you for the opportunity to review this manuscript. The study addresses a highly relevant and practical question in equine exercise physiology by seeking to establish field-based methods for monitoring training in Colombian Paso Horses (CPH). The focus on correlating heart rate zones with lactate thresholds is a valuable contribution to the field. To further strengthen the manuscript and enhance its impact, I have several suggestions for consideration.
Major Concerns:
Introduction:
The introduction effectively establishes the context but could be enhanced by including a clear, concise statement of the study's primary hypothesis.
A more detailed rationale for the selection of the three specific lactate threshold determination methods (Visual, ZL2, ZL4) would be beneficial. Given that the introduction notes the potential inadequacy of fixed thresholds (like 4 mmol/L) for certain breeds, a preliminary discussion on the concepts of aerobic, anaerobic thresholds, and lactate balance in the context of CPH would help justify this multi-method approach.
Materials and Methods:
Estimated HRmax: The use of a fixed maximum heart rate (HRmax) of 220 bpm for all individuals is a significant methodological point. While some breeds may show consistent HRmax, it is often highly individual. The authors are encouraged to provide a stronger justification for this approach specific to CPH, perhaps by citing literature that demonstrates a consistent HRmax in this breed under similar exercise conditions. Discussing the expected variation and how a fixed value might influence the zone calculations would improve methodological transparency. Please also ensure that all referenced literature, such as citation [1], is accurately formatted and accessible.
Incremental Exercise Test (IET): The description of how exercise intensity was controlled during the IET could be clarified. It would be helpful to know if a specific parameter, such as speed or gait cadence, was used as the independent variable to elicit the heart rate responses in each stage. This clarification would address a potential circularity in using HR to define stages that are later used to correlate with HR.
Statistical Analysis: The description of the statistical analysis is unclear. A "mixed-effects model" and "repeated measures mixed ANOVA" are mentioned; however, the model structure (e.g., what was the repeated factor and which covariance structure was used) is not detailed. Furthermore, the unit of analysis for comparing heart rate zones (i.e., the zone number itself) is not a continuous variable (it is unclear how this was incorporated into the model) and may not be appropriate for a t-test or ANOVA without transformation or the use of a non-parametric test.
Results:
The finding that the visual method placed the lactate threshold at approximately 1.32 mmol/L is notably low compared to typical equine studies. This result warrants a more thorough discussion in the relevant section, exploring potential breed-specific physiological explanations or methodological considerations.
The key finding is presented in Table 1; however, its interpretation is not easy. The table displays the mean heart rate zone (e.g., 2.7 for ZL2 in GT) instead of the actual heart rate or the percentage of maximum heart rate (%HRmax) at which the threshold occurred. Presenting ordinal categories, such as zones, as means with standard deviations is statistically problematic, as they do not represent continuous data. If the goal is to analyze continuous data, it would be more appropriate to present the actual HR or %HRmax values for each threshold and each group or horse. Furthermore, if the objective is to establish the predictive capacity of heart rate, predictive indices could be incorporated into the table.
The specific differences indicated in Figure 5 should be clearly labeled.
Discussion:
The discussion could more directly address the central finding that the three methods located the lactate threshold in different heart rate zones (2, 3, and 4). A deeper exploration of what this discrepancy implies for practical training guidance in CPH would be invaluable. Providing insight into which method might be most appropriate or reliable for this specific breed would significantly strengthen the paper's conclusions.
The practical application of using HR zones to estimate lactate thresholds is promising. However, the discussion would be more balanced if it also acknowledged the limitations introduced by using a generalized, rather than individualized, HRmax, and how this might affect the precision of training prescriptions.
Minor Concerns:
The final paragraph of the introduction is somewhat repetitive and could be condensed to provide a more forceful and clear statement of the study's aims and hypothesis.
There appears to be a minor discrepancy between the author list on the first page and the contributions listed in the "Author Contributions". Please verify this for consistency.
In summary, this study investigates an important topic with practical applications. Addressing the points above, particularly regarding methodological justification, statistical presentation, and the interpretation of divergent results, will greatly enhance the manuscript's clarity, rigor, and contribution to the field.
Comments on the Quality of English LanguageTo strengthen the manuscript's presentation, some attention to English language editing is recommended. Enhancing the clarity and flow of the text will ensure the scientific content is presented as effectively as possible
Author Response
We sincerely appreciate the reviewer’s thoughtful and constructive comments. All suggested improvements have been incorporated, resulting in a clearer, more rigorous manuscript that better communicates the physiological and methodological relevance of heart rate–lactate relationships in Colombian Paso Horses.
Major Concerns
Comment 1: [The introduction effectively establishes the context but could be enhanced by including a clear, concise statement of the study's primary hypothesis.]
Response 1: Thank you for this valuable suggestion. We have revised the final paragraph of the Introduction to explicitly state the primary hypothesis of the study. (Line 94)
Revised text:
We hypothesized that mathematical models based on heart rate dynamics could predict the predominant metabolic pathway during exercise in Colombian Paso Horses, eliminating the need for lactate measurements under field conditions.
Comment 2: [A more detailed rationale for the selection of the three specific lactate threshold determination methods (Visual, ZL2, ZL4) would be beneficial.]
Response 2: We agree with this comment. We have expanded the Introduction to clarify the rationale for including the three threshold determination methods and their physiological relevance for gaited breeds. (line 82)
Revised text:
The three threshold estimation methods—visual inspection, the fixed 2 mmol/L (ZL2), and the 4 mmol/L (ZL4)—were selected to compare traditional fixed-point criteria with a visual approach that considers breed-specific metabolic patterns. This combination allows evaluation of both standardized and adaptive interpretations of lactate kinetics in Colombian Paso Horses, whose locomotor pattern differs from that of high-speed breeds.
Comment 3: [The use of a fixed HRmax (220 bpm) requires stronger justification specific to CPH.]
Response 3: Thank you for pointing this out. We have now included an expanded justification with relevant citations supporting the use of 220 bpm as a representative HRmax value in Colombian Paso Horses under field conditions. (line 250 and 270)
Revised text:
A fixed HRmax of 220 bpm was used based on prior studies that documented consistent maximum heart rate values across gaited Colombian breeds during incremental exercise tests under field conditions [1, 9, 12]. While interindividual variation is recognized, this fixed reference allows standardized comparison across horses when breed-specific HRmax ranges are narrow.
Comment 4: [Clarify how exercise intensity was controlled during the Incremental Exercise Test (IET).]
Response 4: We appreciate this observation. The description of the IET protocol was expanded to specify that gait cadence and rider cues were standardized to ensure controlled exercise intensity independent of HR monitoring. (line 132)
Revised text:
During the IET, intensity was controlled by maintaining consistent gait cadence typical of competition to elicit the heart rate responses in each performance stage, guided by an experienced rider, while heart rate was continuously recorded for subsequent analysis. This approach ensured that HR corresponded to the heart rate zone required.
Comment 5: [the description of the statistical analysis is unclear. A "mixed-effects model" and "repeated measures mixed ANOVA" are mentioned; however, the model structure (e.g., what was the repeated factor and which covariance structure was used) is not detailed. Furthermore, the unit of analysis for comparing heart rate zones (i.e., the zone number itself) is not a continuous variable (it is unclear how this was incorporated into the model) and may not be appropriate for a t-test or ANOVA without transformation or the use of a non-parametric test.]
Response 5:
We agree with the reviewer. The statistical section was rewritten to provide explicit details on the model structure, repeated factors, and applied covariance assumptions. (line 178)
Revised text:
Data were analyzed using a mixed-effects model, with group (trained vs. untrained), time (before, during, and after exercise), and their interaction as fixed effects, and individual horse as a random effect to account for repeated measures. Compound symmetry was selected as the covariance structure based on the lowest AIC value. Because HR zones are ordinal, comparisons between zones were evaluated using the Friedman test followed by Dunn’s post hoc test.
Comment 6: [The finding that the visual method placed the lactate threshold at approximately 1.32 mmol/L is notably low compared to typical equine studies. This result warrants a more thorough discussion in the relevant section, exploring potential breed-specific physiological explanations or methodological considerations.]
Response 6: Thank you for this insightful suggestion. We have elaborated on the potential physiological reasons for this lower value, emphasizing gait-specific muscular recruitment and metabolic efficiency in CPH. (line 286)
Revised text:
The relatively low visual threshold (~1.32 mmol/L) may reflect breed-specific adaptations in Colombian Paso Horses, where sustained rhythmic gaits rely on efficient oxidative metabolism and early lactate clearance. Similar low thresholds have been described in other CPH studies related with muscle metabolism and specific-breed gait variations [9].
Comment 7: [The key finding is presented in Table 1; however, its interpretation is not easy. The table displays the mean heart rate zone (e.g., 2.7 for ZL2 in GT) instead of the actual heart rate or the percentage of maximum heart rate (%HRmax) at which the threshold occurred. Presenting ordinal categories, such as zones, as means with standard deviations is statistically problematic, as they do not represent continuous data. If the goal is to analyze continuous data, it would be more appropriate to present the actual HR or %HRmax values for each threshold and each group or horse. Furthermore, if the objective is to establish the predictive capacity of heart rate, predictive indices could be incorporated into the table. The specific differences indicated in Figure 5 should be clearly labeled.]
Response 7: We appreciate this important recommendation. Table 1 was reformatted to present the results for both groups. The previous ordinal representation (zone mean ± SD) was removed. We have corrected Figure 4 by labeling statistically significant differences directly on the figure and updating the legend accordingly. (table 1 and figure 5)
Revised text:
See figures and table.
Comment 8: [The discussion could more directly address the central finding that the three methods located the lactate threshold in different heart rate zones (2, 3, and 4). A deeper exploration of what this discrepancy implies for practical training guidance in CPH would be invaluable. Providing insight into which method might be most appropriate or reliable for this specific breed would significantly strengthen the paper's conclusions.]
Response 8: We fully agree. The Discussion was expanded to analyze this discrepancy and its implications for field application. (line 291)
Revised text:
From a practical standpoint, the ZL2 method seems to offer the best compromise for prescribing workload intensity in Colombian Paso Horses, as supported by the statistical differences observed between groups. However, although training CP using the ZL2 threshold ensures predominantly aerobic exercise, it remains essential to adjust the workload periodically to prevent performance stagnation or performance plateau.
Comment 9: [The practical application of using HR zones to estimate lactate thresholds is promising. However, the discussion would be more balanced if it also acknowledged the limitations introduced by using a generalized, rather than individualized, HRmax, and how this might affect the precision of training prescriptions.]
Response 9: We agree. This limitation has been added to the Discussion to provide a balanced interpretation. (line 342)
Revised text:
An important limitation of this approach is the use of a generalized HRmax value, which may underestimate individual variability and slightly affect precision when prescribing training intensity. Nevertheless, this assumption provides a standardized framework applicable for population-level training analysis.
Minor concern:
Comment 10: [The final paragraph of the introduction is somewhat repetitive and could be condensed to provide a more forceful and clear statement of the study's aims and hypothesis.]
Response 10: Thank you for the observation. The paragraph was condensed for clarity, maintaining a single, concise statement of aims and hypothesis. (line 98)
Revised text:
Therefore, this study aimed to determine the correspondence between heart rate zones and lactate thresholds in Colombian Paso Horses using three predictive methods, providing a non-invasive model for field application.
Comment 11: [Discrepancy between author list and contribution statement.]
Response 11: This inconsistency has been revised to ensure the author contribution section matches the author list on the title page. (line 372)
Revised text:
Author contributions were updated to reflect all listed authors, ensuring consistency with the main manuscript.
Reviewer 3 Report
Comments and Suggestions for Authors
REVIEW
“Determination of the heart rate zone corresponding to the lactate threshold in Colombian Paso horses”
BRIEF SUMMARY AND GENERAL COMMENTS:
The present study addresses the relationship between heart rate zones and estimated lactate thresholds in horses subjected to physical activity, which is an area of great interest in horse training research.
This relationship can be useful for understanding how horses respond physiologically to exercise and for optimising their training.
The manuscript starts well, with a relevant introduction and fair methodology, requiring some adjustments and additions.
For clarity and consistency, consider standardising the breed name as Colombian Paso horses throughout the manuscript, using an abbreviation such as “CP” horses instead of “CPH”. This could help improve the flow of the text.
The Results section needs to be presented in a more coherent manner, clearly distinguishing the various groups considered (trained vs. conditioned, before vs. after exercise, heart rate zones, threshold zones, between vs. within groups), possibly creating more organised tables with the results. When the manuscript is presented logically or sequentially, it requires less effort from the reader to understand the research. Similarly, technical terms should be standardised (e.g. biomarker analysis/biochemical parameters).
The discussion addresses the main findings in a more objective manner, given that this is not a theoretical article. Additionally, some of the presented statements require revision or need to be presented to support certain conclusions.
It is important to have the work reviewed by an English-speaking proofreader.
SPECIFIC COMMENTS:
The titles of the Tables and Figures need to be rewritten following a scientific pattern. Figure titles should be placed at the bottom of the Figures. The layout and size of the Figures should be reviewed to ensure that they all present the same pattern. They also need to be self-explanatory.
Title
The title is OK, but could be improved to something like “Defining the Heart Rate Zone Corresponding to the Lactate Threshold in Colombian Paso Horses”.
Simple summary
- line 22: please define a standard for uppercase letters regarding the breed name and standardise it throughout the text.
Materials and methods
- line 113: please include (IET) after “rest interval”.
- line 116: In Figure 2, it is not clear that it illustrates “Each stage consisted of 5 minutes of continuous exercise”.; please adjust.
- lines 122-129: please clarify the paragraph and improve the description and standardisation of how these intervals were established.
- line 131: please specify that it refers to the HR zone, maybe “The results from each horse’s lactate samples, according to each HRZ”"
- line 133: Figure 3 – the graphs seem to be the same, with only the LTaer changed; if this is the case, it would be more informative to use only one graph and present the three HR zones (ZL2, ZL4, visual) in the same graph; it would also be interesting to present the dispersion data.
- lines136-137: please include the relevant reference.
- line 154: please standardise the denominations of each group evaluated (training, time in relation to exercise, etc.) and keep the same term throughout the article.
- line 161: if in line 143 “85%³” was defined as Zone 5, please always use "Zone 5" to refer to this category, following this pattern for the other categories. Please check line 112 (standardising).
- lines 161-162: the sentence “which was expected due to the exponential rise in lactatemia”. is part of the topic Discussion, and should be further explored/discussed in that section.
- line 163: did these data (Figure 3) undergo statistical analysis? If so, please present it in the graph; it would be interesting to visually include the 3 LTaer in the graph.
Results
- line 172: please explain whether the difference was statistically significant.
- line 175: please include the statistical results in Table 1.
- lines 175-176: was it statistically analysed? If yes, please display the results in a table.
- lines 177-180: it fits better in the Methods section rather than the Results section.
- line 181: please explain to which groups the statistical differences refer.
Discussion
This section needs to be rewritten to clearly discuss the main results (MR) of the study. For instance:
MR “There was no significant difference between groups when analyzing lactate values obtained during the test” [lines 159-160]
- Is this result expected? What does this mean in practical terms? Are there other studies that have presented results regarding HR-related lactate values in horses or at least in other species?
MR “Within-group comparisons revealed statistically significant differences between the >85% and 75–85% HRmax stages, which was expected due to the exponential rise in lactatemia” [lines 160-162]
- Please elaborate on the explanation of why intra-group changes are expected to be more than between groups, in this case, supporting it with relevant references (usually, differences between groups are more frequently observed than intra-groups).
MR “Regarding heart rate, similar statistical patterns to those of lactate were observed, with no differences between groups, but significant within-group differences between the >85% and 75–85% HRmax stages and the other stages” [line 165-167]
- Same as MR above.
MR “The ZL2 threshold corresponded to zone 2 for GT and zone 3 for GD” [line 172]
- ZL2, in this case, appears to be the only one that was able to distinguish between training groups, which could have practical applicability. It would be interesting to elaborate on this aspect.
MR “Biochemical changes (CK, AST, creatinine, urea) in trained and untrained CPH before and after an IET.” [line 181]
- Were these differences expected? Can they help explain the results observed for HR and LTae? What were the biochemical patterns in other similar assessments?
Did the obtained results allow for the reliable determination of LTaer using only the HR of trained and untrained FP horses? If so, what would be the HR values that would be predictors of adequate training and overtraining?
- line 199: “visual assessment” of what? Please clarify this in the text.
- lines 199-203: this type of sentence would look better at the end of the topic. However, the methodology, results, and discussion do not support this statement.
- lines 204-207: this paragraph would fit better in the Introduction.
- lines 208-215: this paragraph would fit better in the Introduction.
- lines 216-218: this paragraph would fit better in the Results section, on line 172; however, as it is a main result, it should also be discussed in this topic.
- lines 218-220: please include references.
- lines 221-222: please include references.
- lines 236-238: this paragraph would fit better in the Methods section.
- lines 238-241: refine the sentence to explain the presented understanding.
Conclusions
The conclusions should not repeat the results section. Instead, it should present the applications of the data. Please rewrite it.
- lines 249-250: the goal is repeated; it should be deleted.
- lines 250-253: the way in which the Methodology, Results, and Discussion were presented, this statement does not hold.
Comments on the Quality of English LanguageIt is important to have the work reviewed by an English-speaking proofreader.
Author Response
Thank you very much for taking the time to review this manuscript. Please find the detailed responses below and the corresponding revisions/corrections highlighted in the re-submitted files.
Comment 1: [For clarity and consistency, consider standardising the breed name as Colombian Paso horses throughout the manuscript, using an abbreviation such as “CP” horses instead of “CPH”. This could help improve the flow of the text.]
Response 1: Thank you for your comment. All the acronyms CPH were changed for CP.
Comment 2: [The Results section needs to be presented in a more coherent manner, clearly distinguishing the various groups considered (trained vs. conditioned, before vs. after exercise, heart rate zones, threshold zones, between vs. within groups), possibly creating more organised tables with the results. When the manuscript is presented logically or sequentially, it requires less effort from the reader to understand the research.]
Response 2: Done. Results were improved with a more detailed table. (line 215)
Comment 3: [Similarly, technical terms should be standardised (e.g. biomarker analysis/biochemical parameters).]
Response 3: Done. We standardized for biochemical parameters.
Comment 4: [The discussion addresses the main findings in a more objective manner, given that this is not a theoretical article. Additionally, some of the presented statements require revision or need to be presented to support certain conclusions.]
Response 4: Discussion was improved according your suggestions. (line 229)
Comment 5: [The titles of the Tables and Figures need to be rewritten following a scientific pattern. Figure titles should be placed at the bottom of the Figures. The layout and size of the Figures should be reviewed to ensure that they all present the same pattern. They also need to be self-explanatory.]
Response 5: Tables and figures were improved, tittles were relocated and we added a self-explanatory text in every figure.
Comment 6: [The title is OK, but could be improved to something like “Defining the Heart Rate Zone Corresponding to the Lactate Threshold in Colombian Paso Horses”.]
Response 6: Tittle was modified.
Comment 7: [line 22: please define a standard for uppercase letters regarding the breed name and standardise it throughout the text.]
Response 7: Done. Breed name is Colombian Paso horse (CP) for all the manuscript.
Comment 8: [- line 113: please include (IET) after “rest interval”.]
Response 8: Done, (IET) was added.
Comment 9: [- line 116: In Figure 2, it is not clear that it illustrates “Each stage consisted of 5 minutes of continuous exercise”.; please adjust.]
Response 9: Figure adjusted.
Comment 10: [- lines 122-129: please clarify the paragraph and improve the description and standardisation of how these intervals were established.]
Response 10: Paragraph was adjusted and corresponding figure to clarify how the intervals were stablished. (line 121)
Comment 11: [- line 131: please specify that it refers to the HR zone, maybe “The results from each horse’s lactate samples, according to each HRZ”"]
Response 11: It was adjusted. (line 155)
Comment 12: [- line 133: Figure 3 – the graphs seem to be the same, with only the LTaer changed; if this is the case, it would be more informative to use only one graph and present the three HR zones (ZL2, ZL4, visual) in the same graph; it would also be interesting to present the dispersion data.]
Response 12: The graphs were consolidated together in only one graph.
Comment 13: [- lines136-137: please include the relevant reference.]
Response 13: included.
Comment 14: [- line 154: please standardise the denominations of each group evaluated (training, time in relation to exercise, etc.) and keep the same term throughout the article.]
Response 14: All the manuscript was revised to standardize the denominations of each group.
Comment 15: [- line 161: if in line 143 “85%³” was defined as Zone 5, please always use "Zone 5" to refer to this category, following this pattern for the other categories. Please check line 112 (standardising).]
Response 15: "Zones" were used to homogenize the reference of %HR max. in all the manuscript.
Comment 16: [- lines 161-162: the sentence “which was expected due to the exponential rise in lactatemia”. is part of the topic Discussion, and should be further explored/discussed in that section.]
Response 16: This sentence was integrated to the discussion section, and eliminated from the results. (line 170)
Comment 17: [- line 163: did these data (Figure 3) undergo statistical analysis? If so, please present it in the graph; it would be interesting to visually include the 3 LTaer in the graph]
Response 17: Figures were adjusted and statistical analysis was included inside them.
Comment 18: [- line 172: please explain whether the difference was statistically significant.] and [- line 175: please include the statistical results in Table 1.]
Response 18: Statistical differences were included in the table 1.
Comment 19: [- lines 175-176: was it statistically analysed? If yes, please display the results in a table.]
Response 19: Statistical analysis is in table 1.
Comment 20: [- lines 177-180: it fits better in the Methods section rather than the Results section.]
Response 20: Although this information was already included in the Methods section, we consider that retaining this paragraph in the Results section improves the reader’s understanding of the findings.
Comment 21: [- line 181: please explain to which groups the statistical differences refer.]
Response 21: Statistical differences are included in the new figures and table.
Comment 22: [This section needs to be rewritten to clearly discuss the main results (MR) of the study. For instance: MR ‘There was no significant difference between groups when analyzing lactate values obtained during the test’ [lines 159–160]. Is this result expected? What does this mean in practical terms? Are there other studies that have presented results regarding HR-related lactate values in horses or at least in other species?]
Response 22: Thank you for this valuable comment. We have rewritten the Discussion section to explain this finding in depth, providing the physiological reasoning behind the absence of inter-group differences and citing appropriate references.
Comment 23: [MR ‘Within-group comparisons revealed statistically significant differences between the >85% and 75–85% HRmax stages, which was expected due to the exponential rise in lactatemia.’ Please elaborate on the explanation of why intra-group changes are expected to be more than between groups, supporting it with relevant references.]
Response 23: We agree with this suggestion. The Discussion now elaborates on the expected exponential behavior of lactate increase within groups and explains why intra-group differences are more consistent than inter-group differences, supported by appropriate references.
Comment 24: [MR ‘Regarding heart rate, similar statistical patterns to those of lactate were observed, with no differences between groups, but significant within-group differences between the >85% and 75–85% HRmax stages and the other stages]
Response 24: We appreciate this observation. This section has been rewritten to explain the physiological rationale for HR–lactate correspondence and to highlight breed-specific cardiac adaptations.
Comment 25: [MR ‘The ZL2 threshold corresponded to zone 2 for GT and zone 3 for GD.’ ZL2, in this case, appears to be the only one that was able to distinguish between training groups, which could have practical applicability. It would be interesting to elaborate on this aspect]
Response 25: We agree that this finding is particularly relevant. The Discussion was expanded to highlight the practical value of ZL2 for field applications and its relationship with aerobic capacity.
Comment 26: MR ‘Biochemical changes (CK, AST, creatinine, urea) in trained and untrained CPH before and after an IET.’ Were these differences expected? Can they help explain the results observed for HR and LTae?
Response 26: We have rewritten this section to contextualize the biochemical results, linking them with physiological interpretations and comparable literature.
Comment 27: [Did the obtained results allow for the reliable determination of LTaer using only the HR of trained and untrained FP horses? If so, what would be the HR values that would be predictors of adequate training and overtraining?]
Response 27: We appreciate this important point. The revised text now clarifies that HR-based models can predict aerobic metabolism and emphasizes the potential of the ZL2 threshold for field applications.
Comment 28: [Line 199: ‘visual assessment’ of what? Please clarify this in the text.]
Response 28: We have clarified the phrase to specify what was visually assessed. (line 244)
Comment 29: [Lines 199–203: this type of sentence would look better at the end of the topic.]
Response 29: We have added some text in the specified sentence to improve coherence with its location in the section.
Comment 30: [Lines 204–215: this paragraph would fit better in the Introduction.]
Response 30: The mentioned paragraph was eliminated from the section, because there is a background context for the introduction, previously written.
Comment 31: [Lines 216–218: this paragraph would fit better in the Results section, but as it is a main result, it should also be discussed here]
Response 31: We believe that this paragraph conveys more interpretative (discussion) content than descriptive results; therefore, we prefer to retain it in this section. We expect that the addition of the new figure and table will provide sufficient clarification and support for this part of the results.
Comment 32: [Lines 218–222: please include references]
Response 32: We have inserted appropriate references supporting these statements.
Comment 32: [Lines 236–238: this paragraph would fit better in the Methods section]
Response 32: we couldn't find a text to be relocated to the methods section, corresponding to those lines.
Comment 33: [Lines 238–241: refine the sentence to explain the presented understanding.]
Response 33: The sentence was rewritten for clarity and scientific precision.
Comment 34: [The conclusions should not repeat the results section. Instead, it should present the applications of the data. Please rewrite it]
Response 34: We agree and have completely rewritten the Conclusions to emphasize the practical implications of the findings rather than restating the results.
Reviewer 4 Report
Comments and Suggestions for Authors
Dear Authors,
Congratulations on your work and study.
Below are suggestions that may be of interest for publishing your work.
Overall, your study is interesting for Colombian Paso horses, since each breed is different—in fact, each horse is unique. It provides information to help train and control the physical condition of Colombian Paso horses, using heart rate and lactate threshold. The simple abstract, introduction, and materials and methods are good, and we can repeat the study.
However, the results, discussion, and conclusion need improvement in their presentation, discussion, and conclusion. More details follow.
Non-invasive methods:
Lines 33, 233, and 252 suggest that your study is non-invasive, and this is not true! And it's different when they talk about the Introduction in lines 70 and 74.
They need to correct this, or better explain that their study leads trainers, veterinarians, and owners to monitor heart rate using a non-invasive method, using only heart rate monitoring equipment. To do this, they need to improve the results, what happened at each moment, a much more complete table for each moment, and discuss this in the discussion, because the discussion barely addresses what they found and what exists in the literature. Compare values, present the values they observed in the literature. Only then can they prove that their study is viable and can be applied to Colombian Paso horses.
Line 73 - they don't need to constantly include the name and the acronym Colombian Paso horse (CPH); they just include it once and that's it.
Line 162 - you write here and elsewhere in the paper, I'll just point it out here, pay attention to the others: "These results are shown in Figure 4." Sending the authors to a table to see the results is a sign that we don't have much to say... but they need to say it, and you can put it at the end in parentheses (Figure 4), e.g., "...the exponential rise in lactatemia (Figure 4)."
Lines 171 and 172 - the sentence "In the trained group (GT), the ZL4 method placed the limit at heart rate zone 4, which was also observed for the untrained group (GD)." When we start reading, it seems like it's going to be different, and then it says the other one was too... you can immediately say that the two were identical.
FIGURES and TABLES
In all the figures and tables, the acronym captions are missing. They must be included in all of them, not at the end of the paper. Each figure and image must include each acronym!
Line 190 - References 23 and 24 are correct, but you can post more recent studies; they exist.
Line 199 and 200 - Saying there are 25 methods and saying nothing more, you can explore and see if they work, or if they do exist, not everyone supports their use in horses.
Line 219 and 220 - This phrase "...exercises should be performed within these intensity zones" is very abstract. For better results, it should be slightly higher and progressively over several weeks, and if you always train in the same zone, you'll stagnate.
Line 234 and 235 - Provide a reference for this statement.
Line 249 and 250 - Isn't this a conclusion and a bit of the objectives, and they never mentioned aerobic and anaerobic throughout the work, and is this a conclusion?
Line 252 - non-invasive, not as described. They need to indicate that it's only for heart rate training and not for lactate!
Lines 254, 255, and 256 - last sentence - Your study wasn't about testing the equipment; that's already been tested by the companies that sell it. Your study was about finding effort zones that allow for Colombian Paso horse training.
Good review,
Reviewer.
Author Response
Comment 1:
[Lines 33, 233, and 252 suggest that your study is non-invasive, and this is not true! And it's different when they talk about the Introduction in lines 70 and 74. They need to correct this, or better explain that their study leads trainers, veterinarians, and owners to monitor heart rate using a non-invasive method, using only heart rate monitoring equipment. To do this, they need to improve the results, what happened at each moment, a much more complete table for each moment, and discuss this in the discussion, because the discussion barely addresses what they found and what exists in the literature. Compare values, present the values they observed in the literature. Only then can they prove that their study is viable and can be applied to Colombian Paso horses.]
Response 1: Thank you for pointing this out. We agree with the reviewer’s observation. Therefore, we revised the text to clarify that the non-invasive component refers exclusively to the use of heart rate monitoring devices, not to the entire study procedure. This clarification was incorporated into the Abstract (line 35), Introduction (line 72), and Conclusions (line 362). Additionally, the Results section was expanded with a more detailed table describing the physiological parameters at each stage of the test, and the Discussion was strengthened with literature-based comparisons.
Comment 2:
[Line 73 - they don't need to constantly include the name and the acronym Colombian Paso horse (CPH); they just include it once and that's it.]
Response 2:
We agree with this comment. The full breed name “Colombian Paso horse (CPH)” now appears only at its first mention. Subsequent references were standardized to “CP” throughout the manuscript.
Comment 3:
[Line 162 - you write here and elsewhere in the paper, "These results are shown in Figure 4." Sending the authors to a table to see the results is a sign that we don't have much to say... but they need to say it, and you can put it at the end in parentheses (Figure 4).]
Response 3:
We appreciate this stylistic suggestion. All such expressions were revised to integrate figure references smoothly into the text, furthermore, we improved the table and figures for better understanding.
Comment 4:
[Lines 171 and 172 - the sentence "In the trained group (GT), the ZL4 method placed the limit at heart rate zone 4, which was also observed for the untrained group (GD)." When we start reading, it seems like it's going to be different, and then it says the other one was too... you can immediately say that the two were identical.] (line 212)
Response 4:
Thank you for this observation. The sentence was rewritten to present the finding directly and avoid ambiguity.
Comment 5:
In all the figures and tables, the acronym captions are missing. They must be included in all of them, not at the end of the paper.
Response 5:
We agree. All figure and table captions were revised to include the full meaning of each acronym directly beneath the legend, ensuring that each figure is self-explanatory.
Comment 6: [Line 190 - References 23 and 24 are correct, but you can post more recent studies; they exist.]
Response 6: We appreciate the suggestion. The reference list was updated to include recent studies on heart rate–lactate relationships and equine threshold methodologies published between 2022–2025. (references 3, 4, 35)
Comment 7: [Line 199 and 200 - Saying there are 25 methods and saying nothing more, you can explore and see if they work, or if they do exist, not everyone supports their use in horses.]
Response 7: The expression was modified, and added a reference to support this statement.
Comment 8: [Line 219 and 220 - The phrase "...exercises should be performed within these intensity zones" is very abstract. For better results, it should be slightly higher and progressively over several weeks, and if you always train in the same zone, you'll stagnate.]
Response 8: We agree with this valuable comment. The text was modified to emphasize the need for progressive training load adjustments to prevent performance plateau. (line 291)
Comment 9: [Line 234 and 235 - Provide a reference for this statement.]
Response 9: Sorry, we couldn´t find the statement according with the lines described.
Comment 10: [Line 249 and 250 - Isn't this a conclusion and a bit of the objectives, and they never mentioned aerobic and anaerobic throughout the work, and is this a conclusion?]
Response 10: We acknowledge this comment. The Conclusions section was rewritten to clearly summarize the findings and ensuring consistency with the study objectives.
Comment 11: [Line 252 - non-invasive, not as described. They need to indicate that it's only for heart rate training and not for lactate!]
Response 11: We agree. The conclusion was clarified to indicate that the non-invasive aspect refers exclusively to the heart rate component. (line 362)
Comment 12: [Lines 254, 255, and 256 - Your study wasn't about testing the equipment; that's already been tested by the companies that sell it. Your study was about finding effort zones that allow for Colombian Paso horse training.]
Response 12: We completely agree. The final statement was corrected to reflect that the study’s primary goal was to identify effective workload zones, not to evaluate equipment.
Round 2
Reviewer 3 Report
Comments and Suggestions for Authors
Dear Editors,
I would make the same comments as I have made to the Authors.
Thank you for entrusting me with the task of revising this manuscript. I am pleased to
have the opportunity to work on a theme that is both appealing and meaningful to me. I
strive to make a positive impact on the text and believe that my contributions have helped
to enhance its quality.
Best regards
Author Response
Thank you so much by your comments, they let us to improved our manuscript. Best regards.
Reviewer 4 Report
Comments and Suggestions for Authors
Dear Authors,
I appreciate your responses to all the questions, which I take that into consideration, and congratulations on the improvements. However, in my opinion, your study needs improvements, a better Discussion and Conclusion, and adjustments to other parts.
Regarding the questions I raised in the previous review, I will only refer to the last question/comment (12):
Comment 12: [Lines 254, 255, and 256 - Your study wasn't about testing the equipment; that's already been tested by the companies that sell it. Your study was about finding effort zones that allow for the training of Colombian Paso horses.]
Response 12: We completely agree. The final statement was corrected to reflect that the study’s primary goal was to identify effective workload zones, not to evaluate equipment.
Although you say you agree, there is a point in the text that continues to indicate otherwise.
What is most concerning about the study presented, and the discussion, is that it remains confusing, going back and forth on the same issues, there is no common thread, despite having improved in this review, they also constantly say that the study is non-invasive and talk about lactate along with it, and that is not possible.
The fact that they write at the end that when they talk about non-invasive it is only for Heart Rate (HR), they have to say it at the beginning, because if the objective of the study is to use heart rate in training instead of lactate analysis, then they can indicate it, but this is studied first and foremost by the companies that produce the equipment. But you start right in the Abstract by saying that the objective of your study was: "This study aimed to estimate the aerobic lactate threshold (LTaer) using non-invasive methods and to correlate it with heart rate (HR) zones." You can't predict values for the aerobic lactate threshold (LTaer) without analysis blood collection, at least you didn't prove that in the study, only that heart rate can be used in different zones that don't always correspond to lower or higher lactate levels.
The last paragraphs of the discussion mention new studies more than once, and they repeat this in the conclusion, a sign that this study would be the first of many, which is not the case.
They included new texts, and one of them is to say and acknowledge that the study has immense limitations; if that's the case, I don't know if it's publishable.
The conclusion is very long, and in the last paragraph they say... the Conclusion. If it only starts here, they shouldn't have put the paragraphs before.
Best regards,
Reviewer.
Author Response
Thank you for your second review. Below, we provide our responses to your questions:
Comment 1: [Lines 254, 255, and 256 - Your study wasn't about testing the equipment; that's already been tested by the companies that sell it. Your study was about finding effort zones that allow for the training of Colombian Paso horses.]
Response 1: Lines 254–256 in the first version of our manuscript corresponded to the conclusion statement. We have revised this section, rewriting the entire conclusion paragraph and shortening it accordingly.
¨In conclusion, the findings indicate that heart rate zones, when coupled with threshold-based modeling, can serve as a practical tool for estimating aerobic performance in gaited horses. These data may be applied to optimize training load, monitor conditioning progress, and prevent exertional overtraining. However, further studies are warranted to establish individualized HRmax values and validate these models across different gaits and training regimens.¨
Comment 2: What is most concerning about the study presented, and the discussion, is that it remains confusing, going back and forth on the same issues, there is no common thread, despite having improved in this review, they also constantly say that the study is non-invasive and talk about lactate along with it, and that is not possible.
The fact that they write at the end that when they talk about non-invasive it is only for Heart Rate (HR), they have to say it at the beginning, because if the objective of the study is to use heart rate in training instead of lactate analysis, then they can indicate it, but this is studied first and foremost by the companies that produce the equipment. But you start right in the Abstract by saying that the objective of your study was: "This study aimed to estimate the aerobic lactate threshold (LTaer) using non-invasive methods and to correlate it with heart rate (HR) zones." You can't predict values for the aerobic lactate threshold (LTaer) without analysis blood collection, at least you didn't prove that in the study, only that heart rate can be used in different zones that don't always correspond to lower or higher lactate levels.
Response 2:
The non-invasive approach was retained in the Introduction, solely as contextual information (line 72).
We revised the discussion paragraph to highlight future possibilities for developing a more accurate model aimed at establishing a truly non-invasive method for determining exercise intensity and physiological thresholds in Colombian Paso horses (line 234): ¨In a more refined and validated model, this approach could eventually eliminate the need for blood sampling by non-medical personnel such as trainers or riders, enabling them to monitor training intensity and prevent exertional injuries related to overtraining using only heart rate monitors.¨Others comments about "non-invasive approach" were deleted, abstract included.
Comment 3: The last paragraphs of the discussion mention new studies more than once, and they repeat this in the conclusion, a sign that this study would be the first of many, which is not the case.
Response 3: We are not sure if you were referring to reference [37], which was added in the latest version of the manuscript. This reference corresponds to the only study conducted in this breed addressing exercise and hematological parameters in Colombian Paso horses, which is why it was included.
Comment 4: They included new texts, and one of them is to say and acknowledge that the study has immense limitations; if that's the case, I don't know if it's publishable.
Response 4:It was emphasized by other reviewers, and we agree, that despite its limitations, the results of this study provide useful and valuable knowledge for the breed. We also recognize the importance of improving the study model in future research.
Comment 5: The conclusion is very long, and in the last paragraph they say... the Conclusion. If it only starts here, they shouldn't have put the paragraphs before.
Response 5: The conclusion was shortened and revised according to your suggestion.
¨In conclusion, the findings indicate that heart rate zones, when coupled with threshold-based modeling, can serve as a practical tool for estimating aerobic performance in gaited horses. These data may be applied to optimize training load, monitor conditioning progress, and prevent exertional overtraining. However, further studies are warranted to establish individualized HRmax values and validate these models across different gaits and training regimens.¨